# Coronary Artery Disease in Women: Lessons Learned from Single-Center SPECT Registry and Future Directions for INOCA Patients

**DOI:** 10.3390/medicina58091139

**Published:** 2022-08-23

**Authors:** Barbara Vitola, Karlis Trusinskis, Iveta Mintale, Marika Kalnina, Andrejs Erglis

**Affiliations:** 1Latvian Centre of Cardiology, Pauls Stradins Clinical University Hospital, LV-1002 Riga, Latvia; 2Faculty of Medicine, Riga Stradins University, LV-1007 Riga, Latvia; 3Faculty of Medicine, University of Latvia, LV-1586 Riga, Latvia; 4Department of Radiology, Pauls Stradins Clinical University Hospital, LV-1002 Riga, Latvia

**Keywords:** coronary artery disease, ischemia with nonobstructive coronary artery disease, nuclear imaging, myocardial perfusion imaging, cardiac single-photon emission tomography, women

## Abstract

*Background and objectives*: Myocardial perfusion imaging with cardiac single-photon emission tomography (SPECT) is widely available for the detection of coronary artery disease (CAD) with high diagnostic and prognostic accuracy for women. A large proportion of symptomatic women with true myocardial perfusion defects in SPECT referred to coronary angiography have an absence of CAD—a condition named INOCA (ischemia with nonobstructive CAD). Additionally, the INOCA endotypes are rarely correctly diagnosed, and therefore, no tailored therapy is prescribed. *Materials and methods*: The cardiac SPECT for women was performed from 2018 to 2021. Patients with perfusion defect were analyzed according to further prescribed diagnostic tests used to diagnose CAD. According to the diagnostic criteria, patients with INOCA were selected as candidates for invasive microvascular physiology measurements. The correlation was calculated between SPECT results and clinical characteristics, symptoms, and risk factors. *Results*: A total of 726 women with suspected CAD were analyzed. True myocardial perfusion defects were detected in 125 patients (17.2%). During coronary angiography in 70 (56.0%) women, atherosclerosis in epicardial arteries was not observed. In 17 (20.9%) patients, obstructive CAD was present. Correlation was found between perfusion defect in SPECT and cardiovascular risk factors, including overweight, obesity, arterial hypertension, and dyslipidemia. Women with typical angina were more likely to have INOCA, but with “noncardiac” symptoms—CAD. In total, 68 female patients met three inclusion criteria for INOCA and were selected as candidates for invasive diagnostic testing. *Conclusions*: The created registry proves the important role of cardiac SPECT and great need for the development of invasively detected physiological measurements. The combination of both interventions could significantly change the future directions for INOCA patients, improving treatment strategies and clinical outcomes, especially knowing the number of risk factors and varying clinical presentation. The study will be continued by performing invasive testing of coronary microvascular function to expand the competence about what is known about INOCA patients.

## 1. Introduction

Based on World Health Organization (WHO) cardiovascular mortality rates, Latvia is in a very high-risk region, and nearly every female patient aged more than 40 years and with non-high-density lipoprotein (non-HDL) cholesterol >3 mmol/L has a high or very high 10-year risk of cardiovascular events as calculated by SCORE-2 or SCORE2-OP risk stratification charts [1]. A population-based cross-sectional study of cardiovascular risk factors (CVRs) in Latvia from 2009 to 2019 also revealed that women aged 25–74 years are in a very high CVR group: 61% of women have dyslipidemia, more than 33% have arterial hypertension (AH), and 28% are overweight [2]. The data indicate the necessity for focused diagnostics and the early detection of cardiovascular disease. Considering the burden of risk factors and varying clinical presentation of female patients with known or suspected coronary artery disease (CAD) [1,3], the creation of a nationwide registry focusing on diagnostic imaging approaches is essential.

Cardiac imaging is a leading method in the diagnosis and treatment of ischemic heart disease. The role of any imaging test is to provide information that clarifies the diagnostic process with the goal of improving patients’ symptoms and clinical outcomes. For patients with low to intermediate pretest probability for CAD, noninvasive imaging has a well-established role in the diagnostic process [4,5]. An initial strategy of exercise tests without imaging is appropriate only for patients with a low pretest probability for CAD who can exercise and have a normal resting electrocardiogram (ECG) [6]. In female patients, the prognostic value of exercise tests is relatively low, and the European Society of Cardiology (ESC) Guidelines recommend using imaging methods with visualization of perfusion [7]. Radionuclide myocardial perfusion imaging with cardiac single-photon emission computed tomography (SPECT) remains the most common form of stress imaging in the diagnostics of patients with known or suspected CAD to evaluate myocardial viability, assess prognosis, and estimate the efficacy of therapy [4]. Patients with significant perfusion defects in cardiac SPECT are referred for invasive diagnostics to confirm the diagnosis of CAD.

Typical chest pain has been mainly caused by atherosclerotic obstructive epicardial CAD [8,9]; however, more than half of the patients undergoing invasive coronary angiography (ICA) or cardiac computed tomography angiography (CCTA) to diagnose CAD have nonobstructive CAD [10]. Patients are often diagnosed with INOCA (ischemia with no obstructive CAD) if the perfusion defect is also present [11,12]. Symptomatic patients without obstructive CAD are associated with a poor quality of life, psychological distress, and healthcare costs that are close to those with obstructive CAD [12]. In these patients, coronary functional disturbances are more likely to be involved [11,13]. Although pathogenetic mechanisms are not completely understood, coronary endothelial and/or microvascular dysfunction, diffuse nonobstructive atherosclerosis, and myocardial ischemia have each been implicated, and these mechanisms can overlap [11,14].

Compared with males, female patients are more likely to have angina without significant coronary artery stenosis presenting with similar symptoms, but have a comparable risk of cardiovascular events, and recent studies have consistently shown an increased female prevalence of coronary microvascular dysfunction (CMD) and microvascular angina (MVA) [14,15,16]. Several studies using both noninvasive and invasive approaches for the assessment of coronary physiology have revealed broad data, improving what is known about INOCA, CMD, MVA, and vasospastic angina (VSA) [11,13,14,15,16,17]. The coronary vasomotor disorders international study (COVADIS) group has created diagnostic criteria for MVA, which include the symptoms of myocardial ischemia, absence of obstructive CAD, objective evidence of myocardial ischemia and impaired microvascular function defined by coronary flow reserve (CFR), microvascular spasm, increased microvascular resistance, and/or coronary “slow flow phenomenon” [18,19].

Female patients with an absence of obstructive CAD occasionally remain underdiagnosed; by improving the accurate diagnostic algorithm for INOCA, more patients, especially in high cardiovascular risk countries, could be diagnosed and treated in a timely manner with higher efficiency.

Currently, in Latvia, invasive techniques for the detection of impaired coronary microvascular function are planned; therefore, one of the purposes of this study was to determine the prevalence of INOCA patients and the necessity for more specific diagnostic tools.

## 2. Materials and Methods

### 2.1. Study Design and Population

This was a single-center, observational cohort study. The study enrolled 726 female patients with known or suspected CAD who underwent cardiac SPECT at Pauls Stradins Clinical University Hospital, Latvian Centre of Cardiology from January 2018 to December 2021. All patients were included in the Latvian cardiac SPECT registry. Patients underwent clinical assessments and received standard medical care as determined by attending physicians. Patients did not receive any experimental intervention. The study protocol was in compliance with the ethics guidelines of the Declaration of Helsinki.

### 2.2. Methods

#### 2.2.1. Clinical Characteristics

Five known risk factors were analyzed—arterial hypertension (AH), dyslipidemia (the low-density lipoprotein cholesterol goal defined by physicians according to patients’ individual status), type-2 diabetes mellitus (T2DM), smoking status, and elevated body mass index (BMI > 25.0 kg/m^2^). The following CAD symptoms were investigated: chest pain, dyspnea, tachycardia/palpitations, excessive sweating, fatigue, and anxiety. According to their clinical symptoms, patients were categorized into “patients with chest pain,” “atypical or noncardiac symptoms,” and “asymptomatic” groups.

#### 2.2.2. Performed Diagnostic Tests

In all patients, the exercise test with the bicycle ergometer and cardiac hybrid SPECT/CT (computed tomography) was performed. CT was used for attenuation correction (no contrast media, with ECG-gating, free tidal breathing). The ECG, heart rate, and arterial blood pressure were recorded during each stage of exercise. Only ≥ 85% of the age-predicted maximum heart rate was considered inclusive for the imaging of maximal myocardial hyperemia. A two-head scintillation Gamma camera (Siemens Symbia T6, IL, USA). in a 90° configuration in a 1-day acquisition protocol was used. The injected 99mTc-tetrofosmin (General Electric (GE) Healthcare, Eindhoven, The Netherlands) activity was 250–300 megabecquerels (MBq) for the stress test. The signal acquisition was started 40 min after the injection. The rest imaging was performed on the same day after 2.5 h, when 700–750 MBq was injected, and after a 35–40 min rest, images were acquired.

At image postprocessing, attenuation-corrected images were used as a diagnostic. To increase the specificity and accuracy of SPECT/CT, physicians considered motion artifacts, soft-tissue attenuation artifacts, and artifacts from the gastrointestinal tract that may create perfusion defects but were not considered as true ischemia.

All registry patients were divided into 2 groups according to the cardiac SPECT results: normal myocardial perfusion and true myocardial perfusion defect, which was defined as an area of ischemia ≥10% of the left ventricle myocardium. For patients with signs of ischemia, invasive coronary angiography (ICA) or cardiac computed tomographic angiography (CCTA) was performed, and the obtained results were analyzed. For female patients with already diagnosed, progressive CAD, immediate percutaneous coronary intervention (PCI) was suggested.

Obstructive CAD was defined as >50% diameter reduction or fractional flow reserve (FFR) <0.80 by ICA or CCTA.

The diagnosis of INOCA and suspected MVA was defined according to COVADIS clinical criteria [17] (Table 1). Suspected MVA was diagnosed if symptoms of ischemia were present (criteria 1) with no obstructive CAD (criteria 2) and only (a) objective evidence of myocardial ischemia (criteria 3) or (b) evidence of impaired coronary microvascular function (criteria 4) alone [17]. Definitive MVA was only diagnosed if all four clinical criteria were present for diagnosis. Discharge on optimal medical treatment (OMT) was prescribed if none of the previous diagnoses were detected.

All patients with significant perfusion defects in cardiac SPECT and with an absence of obstructive CAD were included in the INOCA group to find out the approximate number of patients to whom the measurements of intracoronary physiology should be performed. Only after this procedure is it possible to precisely exclude false positive cardiac SPECT.

The correlation between risk factors, CAD symptoms, and cardiac SPECT results was calculated.

### 2.3. Statistical Analysis

All statistical analyses were performed with IBM SPSS Statistics 27.0 (IBM Corp., Armonk, NY, USA). Continuous variables are expressed as the mean ± standard deviation (SD). Continuous variables were compared using the Mann–Whitney U test, and categorical variables were compared using the chi-squared test and Fisher’s exact test. *p* values < 0.05 were considered significant.

## 3. Results

### 3.1. Diagnostic Approaches

The exercise test was found to be positive in 306 (42.1%) patients. True myocardial perfusion defects on cardiac SPECT were detected in 125 patients (17.2%). Examples are shown in Figure 1 and Figure 2.

As a further diagnostic approach in 29 (23.2%) cases, cardiac CCTA was performed. Significant stenosis in epicardial coronary arteries was not found (100%), and obstructive CAD was excluded.

ICA was suggested in 81 (64.8%) cases. Ten (7.9%) patients from the ICA group were censored due to a lack of information. In 41 (50.6%) women, atherosclerosis in the main coronary arteries was not observed, and suspected INOCA as a primary diagnosis was assessed. In 17 (20.9%) patients, significant stenosis in the main coronary arteries was present, and PCI was performed, among whom 10 (58.8%) women were primarily diagnosed. One patient (1.2%) underwent coronary artery bypass surgery (CBP). In nine (11.1%) patients, significant stenosis was not detected, and they were discharged on OMT. In all nine patients, the CAD was already known before SPECT.

Fifteen (12.0%) patients with known and progressive CAD were immediately referred to PCI: eight women underwent PCI (53.4%), one woman underwent CBP (6.6%), and six (40%) patients continued OMT.

In total, 68 (54.4%, 9.4% from all performed SPECTs) female patients met three inclusion criteria for suspected INOCA. Based on our data, we cannot define the exact diagnosis of INOCA endotypes: CMD, MVA, or VSA, according to COVADIS criteria.

All diagnostic approaches are shown in Figure 3.

### 3.2. Clinical Characteristics and Risk Factors

The mean age of the registry patients was 62.6 ± 10.6 years. More than half of them (*n* = 486, 66.9%) were overweight (BMI > 25.0 kg/m^2^), and 33.3% (*n* = 242) were obese (BMI > 30.0 kg/m^2^).

Eighty-seven (11.9%) female patients were current or ex-smokers. Dyslipidemia was present in 614 (84.6%) patients, AH in 535 (73.7%) patients and T2DM in 60 (8.3%) patients. Known CAD was present in 298 (41.0%) patients. A correlation was found between significant perfusion defects in SPECT and the following risk factors: overweight (*p* = 0.015), obesity (*p* = 0.003), AH (*p* = 0.013), and dyslipidemia (*p* = 0.029). All clinical characteristics are summarized in Table 2.

### 3.3. CAD and INOCA Symptom Analysis

The most common presented symptoms in study patients were chest pain (*n* = 487, 67.1%), dyspnea (*n* = 323, 44.5%), tachycardia/palpitations (*n* = 117, 16.1%), fatigue (*n* = 129, 17.8%), headache (*n* = 28, 3.86%), and excessive sweating (*n* = 15, 2.1%). A total of 104 (14.3%) patients were asymptomatic, 487 (67.1%) had typical chest pain, and 135 (18.6%) had atypical symptoms.

In the “asymptomatic” patient group, obstructive CAD was diagnosed in 5 (4.8%) cases, in the “chest pain” group it was diagnosed in 11 (2.3%) cases, and in the “atypical or noncardiac symptom” group it was diagnosed in 16 (11.9%) patients (*p* < 0.001). None of the asymptomatic women had suspected INOCA. INOCA was detected in 54 (43.2%; 7.4% of all) patients in the “chest pain” group and in 14 (11.3%; 1.9% of all) patients in the “atypical or noncardiac symptom” group (*p* = 0.006).

## 4. Discussion

The precise evaluation of CAD symptoms and characteristics in female patients demands performing guideline-directed strategies of treatment at improving cardiovascular disease outcomes. In Latvia, the usage of cardiac nuclear imaging is increasing; however, there are some limiting factors that do not let researchers completely use this diagnostic approach in women with suspected CAD or INOCA. Over 4 years, we performed approximately 730 cardiac SPECTs for women with suspected CAD. The lack of nuclear cardiologists and radiology technicians is the key problem in performing more cardiac SPECTs.

In recent decades, SPECT and SPECT/CT imaging has, unfortunately, showed limited diagnostic utility for assessment of CMD, mostly because of pharmacokinetics of the technecium-99m (99mTc) radiotracers used for SPECT myocardial imaging [20]. However, with the recent development of high-sensitivity cardiac cameras, iodinated rotenone, and solid-state, high-sensitivity cadmium–zinc–telluride detectors (CZT), dynamic SPECT can be used for quantification of myocardial blood flow and, hence, for the assessment of CMD [21]. The feasibility of myocardial blood flow and myocardial flow reserve (MFR) estimation using dynamic SPECT was assessed in the WATERDAY study. They validated dynamic CZT-SPECT (against ^15^O–water positron emission tomography (PET)) as having a high diagnostic value for detecting impaired MFR in patients with stable CAD [22]. PET/CT provides global and regional measurements of perfusion, quantitative myocardial blood flow (MBF) and function, both at stress and rest, in one examination. Quantification of MBF has been extensively validated with PET, and it is the most widely used, non-invasive modality for the clinical assessment of coronary microvascular disease [23]. Advances in technology and new emerging radiotracers will lead to a better understanding of integrative biology. Additionally, developing cardiac SPECT/CT or cardiac PET as a routine imaging method makes it possible to improve the diagnosis of CAD and INOCA.

Moreover, the created Latvian cardiac SPECT registry shows the importance of nuclear diagnostics for such a difficult-to-diagnose group as women, given the wide spectrum of CAD symptoms and relatively low specificity of the exercise test alone.

Among women, exercise-test-induced ST depression in ECG in the absence of CAD has been described in relation to changes in estrogen levels during the menstrual cycle or from menopausal hormone therapy and has been associated with lower diagnostic accuracy when compared to men [4,24,25]. The results from the WOMEN (What is the Optimal Method of ischemia Elucidation in Women) trial demonstrated no superiority of the exercise test with myocardial perfusion over the exercise test alone in low- to intermediate-risk women who were able to exercise [4,26,27]. On the other hand, in women with intermediate to high pretest probability, the diagnostic accuracy of detecting obstructive CAD is better for exercise tests with perfusion (cardiac SPECT) than for exercise tests alone (cardiac SPECT sensitivity 78% (95% CI 72% to 83%) versus the exercise test only sensitivity 61% (95% CI 54% to 68%)) [28]. In our registry, the exercise test before perfusion imaging was found to be positive in 42% of patients, and the test result was not always associated with ischemia in SPECT (*p* = 0.063). These variances emphasize the necessity for cardiac nuclear imaging, especially in patients with noninformative exercise tests and intermediate to high CVR.

The clinical evaluation of symptomatic women is also challenging due to their varying clinical presentation. Furthermore, the use of “typical” angina symptoms in the assessment of female patients with CAD may be confounded due to the transposition of symptoms analyzed in male cohorts [29,30,31]. Female patients more frequently report additional, nonspecific symptoms such as fatigue, sleep disturbance, excessive sweating, and anxiety [29,30,31,32]. The presentation may therefore lead to misdiagnosis, making the transition of these symptoms “noncardiac.” Differences in presentation tend to be more polarized among younger patients than among older individuals [29,31]. A study of more than 10,000 patients indicated that older women (> 65 years old) presented with symptoms comparable to men and had a greater frequency of typical chest pain [31]; however, the female patients referred to cardiac SPECT are usually younger and have a higher ability to exercise. In our cohort, the mean age of female patients was 62 years, and they presented with an extensive range of CAD symptoms, which enhanced the role of nuclear diagnostics in our study population. The variability of symptoms is identified not only as gender-specific, but also between diagnoses. In our study, we demonstrated the correlation between CAD and INOCA symptoms and cardiac SPECT results and found that female patients with typical chest pain were more likely to have suspicions about INOCA, but with atypical or “noncardiac” symptoms—CAD. The balance between the mentioned clinical characteristics and physicians opting to use the most accurate diagnostic method is crucial for the detection of CAD or INOCA.

INOCA can develop in variable clinical settings and can be caused by multiple pathogenic mechanisms [14,19,33,34]. Similar to CAD, INOCA and its endotypes have been associated with cardiovascular risk factors including age, dyslipidemia, AH, and diabetes mellitus, although the prevalence of these conditions in patients with CMD or VSA remains unknown [19]. According to the COVADIS criteria for MVA, all needed diagnostic approaches to fulfill the criteria may take a longer time and expose relatively higher radiation doses for patients [4] coming in for an exact diagnosis. While these criteria are logical and provide a useful structure for thinking about CMD, the need to perform a combined method with both invasive and noninvasive techniques may be one of the factors that has led to the poor uptake of this in routine clinical practice [35]. However, in clinical practice, this algorithm is not performed enough.

In our research cohort, we detected at least 68 (9.4% from all performed SPECTs) INOCA patients to whom invasive microvascular physiology testing should be performed to precisely interpret INOCA endotypes such as endothelial-dependent or endothelial-independent CMD and VSA. In our study, all patients with significant perfusion defects in cardiac SPECT and with an absence of obstructive CAD were included in the INOCA group to find out the approximate number of patients to whom the measurements of intracoronary physiology should be performed. After this procedure, it was possible to precisely exclude false positive cardiac SPECT, especially in female patients, due to attenuation from breast or general adipose tissue in those with high BMI [4]. Additionally, the technical reasons, such as attenuation correction, could significantly decrease the number of false-positive SPECTs. Meanwhile, the false-negative cardiac SPECT result in women should be taken into consideration, because of partial volume effects, which are amplified in smaller left ventricles in a female population [4].

The reason for additional testing of coronary vascular function during ICA is relevant due to diagnosis, prognosis, and treatment indications. In a symptomatic patient with INOCA, coronary angiography is considered incomplete without additional invasive tests of coronary vascular dysfunction [7,36,37,38]. The detection of CMD as a mechanism or cause of myocardial ischemia provides new prognostic information, allowing physicians to select optimal guideline-directed therapy.

In 2018, the Journal of American College of Cardiology (JACC) published results from the “CorMicA trial”, which aimed to test whether an interventional diagnostic procedure linked to stratified medicine improves health status in patients with INOCA. The results indicated that medical therapy, including an interventional diagnostic test to confirm diagnosis, is routinely appropriate and improves angina symptoms in patients with no obstructive CAD [39]. 

The two main cornerstones of invasive microvascular testing to prove CMD are: measurements of coronary flow reserve (CFR), and index of microvascular resistance (IMR). Both diagnostic approaches include coronary thermodilution using a pressure–temperature sensor guidewire or a Doppler wire technique [36,38,40,41]. The usual approach to induce steady-state hyperemia is by use of intravenous adenosine. It activates vascular A2 receptors, leading to predominantly nonendothelium-dependent vasodilation [42]. CFR is calculated using thermodilution as resting mean transit time divided by hyperemic mean transit time. Consecutively, the IMR is calculated as the product of distal coronary pressure at maximal hyperemia multiplied by the hyperemic mean transit time [36].

In the diagnosis of VSA, the most established approach for vasoreactivity testing is by intracoronary infusion of acetylcholine [22]. When microvascular spasm develops, coronary flow temporarily reduces in the absence of epicardial coronary artery spasm, which means that the diameter of the coronary diameter is maintained in association with transient reduction in flow (TIMI) flow grade ≤ 2), while the patient generally experiences chest pain in association with ischemic changes on electrocardiography [24,36].

Based on our pilot study results, we will continue this work by performing invasive testing of coronary microvascular function and physiology, including measurements of CFR and IMR in the Latvian Centre of Cardiology, to expand the knowledge about what is known about INOCA patients.

The treatment of INOCA patients has been discussed; however, no specific guidelines are available due to a lack of randomized clinical trials. As a consequence, these patients continue to experience recurrent angina with impaired quality of life, leading to repeated hospitalizations, unnecessary coronary angiography, and adverse cardiovascular outcomes in the short and long term [14]. Previously considered as “low risk” [43], the newest evidence documents a higher-than-expected risk for major adverse cardiac events (MACEs), defined as death, myocardial infarction, stroke, or hospitalization for chest pain or heart failure, compared to asymptomatic subjects [15,40,44,45,46]. In 2021, in the United States of America, a new randomized clinical trial called WARRIOR (Women’s ischemiA tRial to Reduce events In non-ObstRuctive CAD) was started to investigate the strategy of intensive medical treatment versus usual care in women with evidence of INOCA for the reduction of MACE [47]. Currently, treatment options are lifestyle and risk factor management and antianginal medication, including beta blockers, calcium-channel blockers, angiotensin-converting enzyme inhibitors, nitrates, and statins [14]. To prescribe this medication for patients, the precise diagnostics and differentiation of clinical symptoms between angina and noncardiac complaints are relevant.

Precise CAD and INOCA diagnostics in combination with invasive and noninvasive methods will allow more targeted therapy for this patient group.

## 5. Conclusions

The created Latvian cardiac SPECT registry proves the necessity and important role of myocardial perfusion imaging with cardiac SPECT in women with CAD or INOCA. More usage of nuclear diagnostics and its new technological advancements in Latvia is preferable to improve the knowledge of early predictors of INOCA or CAD in cardiac SPECT.

Data from this pilot study provide a great need for the development of invasively detected physiological measurements to define the diagnosis of INOCA and its endotypes—microvascular angina, previously known as “Cardiac Syndrome X”, and vasospastic angina—as well as to precisely exclude the false-positive SPECTs, especially within the female population.

The performance of invasive diagnostic methods in combination with cardiac SPECT could significantly change the future directions for INOCA patients, improving treatment strategies and clinical outcomes.

## Figures and Tables

**Figure 1 medicina-58-01139-f001:**
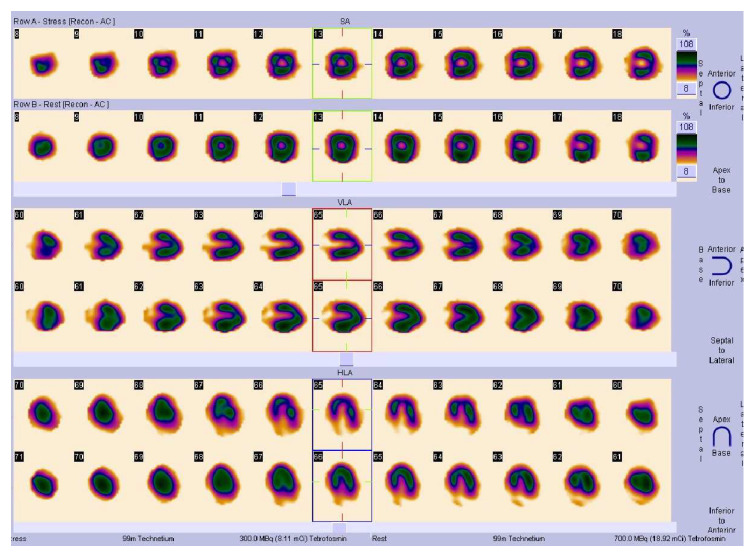
SPECT/CT images. Axial, sagittal, and coronal perfusion images showing stress-induced transmural perfusion defect in apical anterior and lateral walls in visual assessment. PACS system Sectra Workstation IDS7 Version 21.2, IHE, Chicago, IL, USA.

**Figure 2 medicina-58-01139-f002:**
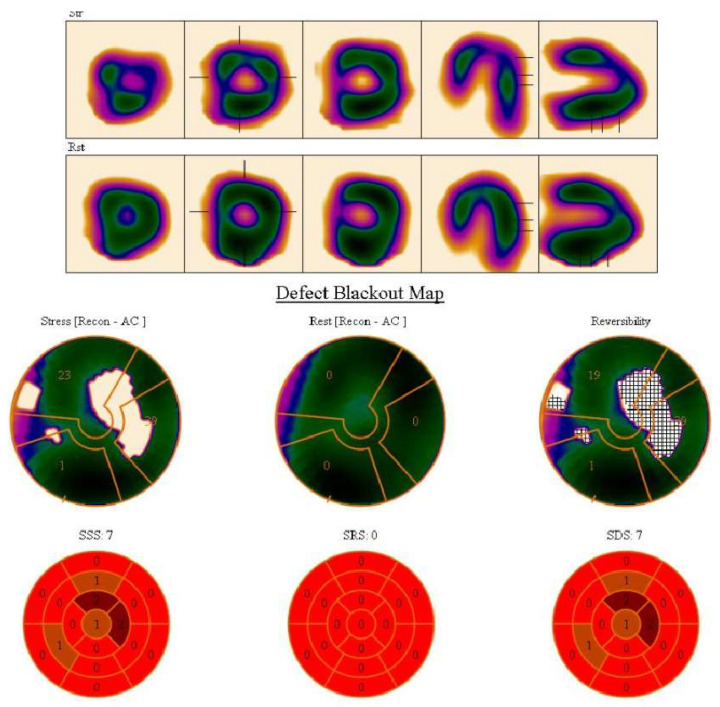
SPECT/CT images. SSS: summed stress score, SRS: summed rest score, SDS: summed difference score. Stress, rest and perfusion reversibility polar maps showing reversible perfusion defect with semiquantitative summed difference score 7, representing low to intermediate risk for a cardiac event. PACS system Sectra Workstation IDS7 Version 21.2, IHE, Chicago, IL, USA.

**Figure 3 medicina-58-01139-f003:**
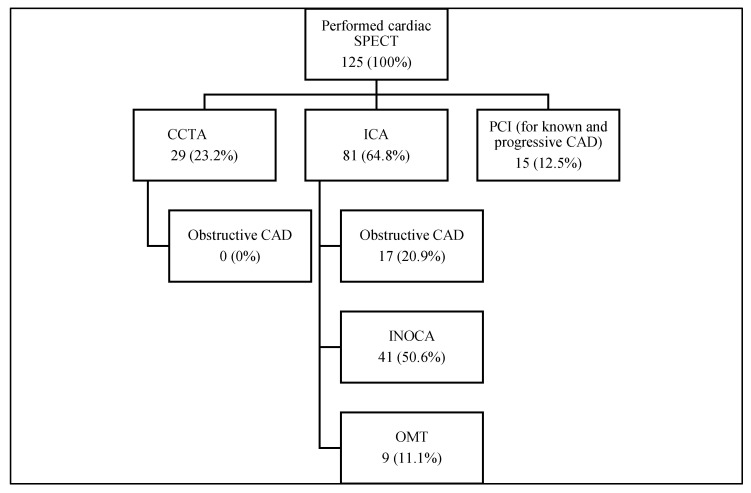
Diagnostic approaches suggested after cardiac SPECT and results. Figure legend: SPECT: single-photon emission computed tomography, CCTA: computed coronary angiography, ICA: invasive coronary angiography, CAD: coronary artery disease, OMT: optimal medical treatment, PCI: percutaneous coronary intervention.

**Table 1 medicina-58-01139-t001:** Clinical criteria for INOCA endotype—microvascular angina (MVA). Updated from COVADIS Group [11].

**1.**	**Symptoms of myocardial ischemia:**
	Effort and/or rest angina
	Angina equivalents (i.e., shortness of breath)
**2.**	**Absence of obstructive CAD by:**
	Computed coronary angiography
	Invasive coronary angiography
**3.**	**Objective evidence of myocardial ischemia:**
	Ischemic ECG changes during an episode of chest pain
	Stress induced chest pain and/or ischemic ECG changes in the presence of absence of transient/reversible abnormal myocardial perfusion and/or wall motion abnormality
**4.**	**Evidence of impaired coronary microvascular function:**
	Impaired coronary flow reserve (cutoff values depending on methodology use between <2.0 and <2.5)
	Coronary microvascular spasm, defined as reproduction of symptoms, ischemic ECG shifts but no epicardial spasm during acetylcholine testing
	Abnormal coronary microvascular resistance indices (e.g., IMR > 25)
	Coronary slow flow phenomenon, defined as TIMI frame count > 25

Table legend: INOCA: ischemia with non-obstructive coronary artery disease, COVADIS: coronary vasomotor disorders study group, ECG: electrocardiogram, CAD: coronary artery disease, IMR: index of microcirculatory resistance, TIMI: thrombolysis in myocardial infarction.

**Table 2 medicina-58-01139-t002:** Clinical characteristics and risk factors for female patients referred to cardiac SPECT.

	*N*, % from All SPECTs	*N*, % from SPECTs with Perfusion Defect	95% CI	*p* Value
Age, years ± SD	62.6 ± 10.6	60.1 ± 10.4	(58.26–61.94)	0.004
Body mass index (BMI), kg/m^2^ ± SD	28.1 ± 5.5	27.5 ± 5.6	(26.51–28.49)	0.361
- Overweight (BMI > 25.0 kg/m^2^)	486 (66.9%)	72 (57.6%)	(0.48–0.66)	0.015
- Obese (BMI > 30.0 kg/m^2^)	242 (33.3%)	33 (26.4%)	(0.18–0.35)	0.003
Current or ex-smokers	87 (11.9%)	19 (15.2%)	(0.09–0.22)	0.286
Dyslipidemia	614 (84.6%)	114 (91.2%)	(0.84–0.95)	0.029
Arterial hypertension	535 (73.7%)	81 (64.8%)	(0.55–0.73)	0.013
Type 2 diabetes	60 (8.3%)	10 (8.0%)	(0.03–0.14)	0.951
Known CAD	298 (41.0%)	43 (34.4%)	(0.26–0.43)	0.119
Positive exercise test	306 (42.1%)	62 (49.6%)	(0.40–0.58)	0.063

Table legend: BMI: body mass index, CAD: coronary artery disease, SPECT: single-photon emission computed tomography, SD: standard deviation.

## Data Availability

The data presented in this study are available from the corresponding author on reasonable request.

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
