# Peer review of "Coronary Artery Disease in Women: Lessons Learned from Single-Center SPECT Registry and Future Directions for INOCA Patients"

_medicina, 2022, doi:10.3390/medicina58091139_

Round 1

Reviewer 1 Report

Dear Editor, 

I had a great pleasure to read an article entitled: "CORONARY ARTERY DISEASE IN WOMEN: LESSONS 2 LEARNED FROM SINGLE CENTRE SPECT REGISTRY AND 3 FUTURE DIRECTIONS FOR INOCA PATIENTS"

In general I have no major issues concerning that article, but I would like to advise: 

1) line 134: ...myocardial mass... - is this right ?

2) I would suggest to find preditors of >10% ischemia in SPECT and coronary angiography and both. 

3) I wolud suggest to expose that, false negative spect in terms of the lack of significant stenoses in coronary arteies in angiography does not mean that really false negative result due to the possible impairment of microvascular circulation 

Author Response

Dear, reviewer! I am grateful to read good words, advice and comments about our study and this article. We improved the following points, mentioned in your suggestions:

  1. We improved the previously mentioned term “left ventricle myocardium mass” to – “left ventricle myocardium” which is more accurate and right.
  2. In our research cohort the predictors of significant perfusion defect in cardiac SPECT as well as of coronary angiography result will be defined and results published in next paper after the performance of intracoronary physiology measurements of suspected INOCA patients. This design is preferable due to possibly false positive SPECTs, which could be excluded only after invasive diagnostic procedure and confirmation of exact diagnosis, like coronary microvascular dysfunction or vasospastic angina. After these interventions also the differences between women and men; INOCA and obstructive CAD; and between INOCA endotypes will be clearly defined. The future of Latvian Cardiac SPECT registry is to improve the diagnostic accuracy of cardiac SPECT excluding false positive results. This plan also includes the detection of INOCA patient’s phenotype and predictors of perfusion defect in nuclear tests.
  3. According to the sensitivity and specificity of cardiac SPECT we extended our work with following paragraph in Discussion section about false negative and false positive results.

“In our study all patients with significant perfusion defect in cardiac SPECT and with an absence of obstructive CAD were included into INOCA group to find out the approximate number of patients to whom the measurements of intracoronary physiology should be performed. After this procedure it is possible to precisely exclude false positive cardiac SPECT, especially in women patients due to attenuation from breast or general adipose tissue in those with high BMI [4, 46]. Also, the technical reasons like attenuation correction could significantly decrease the amount of false positive SPECTs. Meanwhile, the false negative cardiac SPECT result in women should be taken into consideration, because of partial volume effects, which are amplified in smaller left ventricles [4.]”

Thank you for your time reading our work and help you gave us to improve and make our article better! 

Reviewer 2 Report

The paper is interesting, well written and original, the text is clear and easy to read. It focuses on a particular topic. I suggest shortening the conclusion to make it more clear. The conclusion should be improved with a clear distinction between INOCA, X syndrome and False positive

Moreover, the authors should stress the concept of "false positive" in the spect; from how the work has been described, every positive spect with negative angiography has been inserted in the INOCA arm, but we know that, especially in women, the SPECT can give a large percentage of false positives, I consider it a duty for the authors to clarify this concept.

Rewrite the sentences line 67

Author Response

Dear, reviewer! I am grateful to read good words, advice and comments about our study and this article. We improved the following points:  

  1. After your suggestion we have updated the conclusion section of our article, focusing on . 

“The created Latvian cardiac SPECT registry proves the necessity and important role of myocardial perfusion imaging with cardiac SPECT in women with CAD or INOCA. More usage of nuclear diagnostics and its new technological advancements in Latvia is preferable to improve the knowledge of early predictors of INOCA or CAD in cardiac SPECT.

Data from this pilot study provide a great need for the development of invasively detected physiological measurements to define the diagnosis of INOCA and its endotypes – microvascular angina, previously known as “Cardiac Syndrome X” and vasospastic angina as well as to precisely exclude the false positive SPECTs especially in women population.

The performance of invasive diagnostic methods in combination with cardiac SPECT could significantly change the future directions for INOCA patients improving treatment strategies and clinical outcomes.” 

  1. According to the false positive cardiac SPECTs we extended our work with following paragraph about both - false negative and false positive results. Also, improved this in Methods section.

“In our study all patients with significant perfusion defect in cardiac SPECT and with an absence of obstructive CAD were included into INOCA group to find out the approximate number of patients to whom the measurements of intracoronary physiology should be performed. After this procedure it is possible to precisely exclude false positive cardiac SPECT, especially in women patients due to attenuation from breast or general adipose tissue in those with high BMI [4, 46]. Also, the technical reasons like attenuation correction could significantly decrease the amount of false positive SPECTs. Meanwhile, the false negative cardiac SPECT result in women should be taken into consideration, because of partial volume effects, which are amplified in smaller left ventricles in women population [4.]”

3. We have updated the mentioned sentence with abbreviations. 

Thank you for your time for reading our work. We are glad for the help you gave us to improve and to make our article better!

Reviewer 3 Report

After reading the manuscript, I would like to recommend an acceptance for this paper. The authors can improve the quality as comments below.

1. Please rewrite the abstract section. You should focus on the method, main findings and significance of this study. Please add a future perspective for the conclusion section.

2. Please insert error bars in tables

3. Reference style should be checked. 

4. English language should be polished.

Thank you!

Author Response

Dear, reviewer! I am grateful to read good words, advice and comments about our study and this article. Also, thank You for recommending our work for acceptance. We improved the following points:

  1. We rewrited the abstract section, focusing more on the methods used and conclusions as well as future perspective for INOCA patients.
  2. We have inserted the error bars in table.
  3. The reference style is checked according to the MDPI Guidelines.
  4. The English language was previously checked in “American Journal Experts” https://www.aje.com . Please see the certificate in attached files (the Title of the article was changed a little after the editing service in June 15).

Thank you for your time for reading our work and help you have given us to improve and make our article better! We were glad to read the recommendation for acceptance from you.  

Round 2

Reviewer 2 Report

my answers are been addressed